# China's GDP forecasting using Long Short Term Memory Recurrent Neural Network and Hidden Markov Model

**Junhuan Zhang**[1,2]*, **Jiaqi Wen**[3], **Zhen Yang**[1]

**1** School of Economics and Management, Beihang University, Beijing, China, **2** Key Laboratory of Complex System Analysis, Management and Decision (Beihang University), Ministry of Education, Beijing, China, **3** School of Computer Science, University of Technology Sydney, Sydney, Australia

* junhuan_zhang@buaa.edu.cn, junhuan.zhang@gmail.com

**Data Availability Statement:** Data are available from China's National Bureau of Statistics http://www.stats.gov.cn/, which is accessible to anyone. This website provides a free search engine (https://data.stats.gov.cn/) for National data in China.

## Abstract

This paper presents a Long Short Term Memory Recurrent Neural Network and Hidden Markov Model (LSTM-HMM) to predict China's Gross Domestic Product (GDP) fluctuation state within a rolling time window. We compare the predictive power of LSTM-HMM with other dynamic forecast systems within different time windows, which involves the Hidden Markov Model (HMM), Gaussian Mixture Model-Hidden Markov Model (GMM-HMM) and LSTM-HMM with an input of monthly Consumer Price Index (CPI) or quarterly CPI within 4-year, 6-year, 8-year and 10-year time window. These forecasting models employed in our empirical analysis share the basic HMM structure but differ in the generation of observable CPI fluctuation states. Our forecasting results suggest that (1) among all the models, LSTM-HMM generally performs better than the other models; (2) the model performance can be improved when model input transforms from quarterly to monthly; (3) among all the time windows, models within 10-year time window have better overall performance; (4) within 10-year time window, the LSTM-HMM, with either quarterly or monthly input, has the best accuracy and consistency.

## Introduction

Gross Domestic Product (GDP) is a key indicator of economic growth, which measures the total value of goods and services produced within a country in a year, not including its income from investments in other countries. There are considerable literatures on economic prediction in financial market that utilises Hidden Markov Model (HMM) [1, 2] and Long Short-Term Memory (LSTM) recurrent neural network [3, 4]. However, there isn't any study on the application of LSTM-HMM in GDP forecast that combines the potentials of these two models, which motivates our research on the China's GDP forecast using LSTM-HMM considering the relationship between inflation and economic growth.

The dynamic relationship between inflation and economic growth is the basis for the macro-control, and these economic indicators are interrelated, providing reference for GDP trend prediction. There are many researches on the complex relationship between inflation

**Funding:** This study was supported by the National Natural Science Foundation of China (71801008) and Beihang University (KG16156301) through grants awarded to JZ. The funders had no role in study design, data collection and analysis, decision to publish, or preparation of the manuscript.

**Competing interests:** The authors have declared that no competing interests exist.

rate and economic growth including the nonlinear effects and threshold effects. Inflation tends to show possibility of nonlinear effects on economic growth and there is a significant structural break in the function that relates economic growth to inflation [5]. In middle-income and low-income countries, higher inflation is associated with moderate gains in gross domestic product growth up to a roughly 15–18 percent inflation threshold [6]. Also in China, the inflation threshold effect is also highly significant and robust [7], while the synchronization between business and inflation cycles is obvious across Chinese province as China's economy has been liberalized and modernized [8]. Among these researches, Consumer Price Index (CPI) has been utilised as the indicator of inflation rate [5, 7], while Gross Domestic Product (GDP) is also a key indicator economic growth [5, 8].

Threshold models have been widely used in current econometric practice where split samples are classified based on the value of an observed variable-whether or not it exceeds some threshold [9]. The estimation of the threshold is complex as it is typically unknown in econometric practice. A theory of estimation and inference is fairly well developed for linear models with exogenous regressors [10–12], which explicitly exclude the presence of endogenous variables. However, this has been an impediment to empirical application, thus an instrumental variable estimation of a threshold model has been introduced to solve the problems of endogenous variables [13], where an estimator and a theory of inference for linear models with endogenous variables and an exogenous threshold variable have been developed.

Hidden Markov model (HMM) is an effective method to monitor the business cycle and even growth cycle. There are researches on the usefulness of HMM in economic forecast. A multivariate qualitative Hidden Markov Model can measure the probability of being in an accelerating or decelerating phase of French economic activity, proving HMM a useful method of applying business cycle turning point in economic forecast [1]. HMM also shows usefulness in interpreting American market moves to forewarn economic downturns [2], where the financial markets have rarely failed to detect economic turning points.

Long Short-Term Memory (LSTM) is a type of recurrent neural network that works better on tasks involving long time lags by bridging huge time lags between relevant input events [14], which has emerged as an effective and scalable model for time-series prediction. LSTM is applied to CPI prediction in Indonesia with multivariate input [3]. LSTM is also utilised to the optimal hedging in the presence of market frictions [4], which shows usefulness in the empirical analysis of real option markets.

Motivated by the usefulness of HMM and LSTM in economic forecast and the sparsity of literatures on the application of LSTM-HMM in GDP forecast, this paper establishes a LSTM-HMM to predict GDP fluctuation states. We further compare the predictive power of LSTM-HMM with HMM and GMM-HMM using monthly CPI or quarterly CPI within different time windows. There are three innovation points. First, in LSTM-HMM, we innovatively utilise LSTM to predict real-time CPI fluctuation states while feeding this prediction into the forecast of real-time GDP fluctuation states. Second, we select the inflation indicator CPI as the model input and utilise available monthly and quarterly CPI respectively in the LSTM-HMM. The GDP fluctuation states are determined based on the turning points of GDP growth rate estimated by a double-threshold auto-regression model, while the CPI fluctuation states are identified through the instrumental variable estimation of s threshold model based on GDP growth rate and CPI growth rate. Third, we find from the empirical analysis of China's GDP fluctuation states that among all the time windows of rolling prediction, models within 10-year time window have better overall performance in accuracy and consistence and LSTM-HMM generally has good precision. The overall performance can generally be improved as the model input transforms from quarterly to monthly.

This paper is organized as follows. The econometric model and estimation method we employ describes the econometric model and the estimation method. Empirical analysis analyzes the economic indicators involved in empirical analysis, compares the predictive results of different models with an input of quarterly CPI or monthly CPI within different time windows. Conclusion presents the conclusion.

## The econometric model and estimation method we employ

We build a LSTM-HMM structure to predict GDP fluctuation states and introduce other dynamic forecast systems including HMM and GMM-HMM to compare the prediction accuracy. All these models are derived from the benchmark HMM, and they differ in the methods of classifying of CPI fluctuation states. The structures of these models are shown below.

### The benchmark model: HMM

HMM is an effective method to monitor the business cycle and even growth cycle. [1] find the usefulness of HMM to measure the probability of a business cycle turning point of France, therefore we select HMM to predict China's economic activity. Our primary objective is to evaluate the forecasting performance of the HMM employed in this paper. The HMM we build is shown in Fig 1.

HMM is a standard Markov process augmented by a set of measurable states and several probabilistic relations between those states and the hidden states [15]. As we can see from Fig 1, GDP fluctuation state $s$ follows a first order Markov Chain with CPI fluctuation state $v$ observed at each time point, where state transition matrix $A$ is the probabilistic relations between $s$ and the emission probability matrix $B$ stands for the probabilistic relations between $s$ and $v$. We obtain the CPI fluctuation states based on the latest observable information of CPI series in the forecast of GDP fluctuation.

To better illustrate our model, let us define the notations for the HMM framework accurately.

We set $s$ as a hidden variable named GDP fluctuation state, which represents the state of GDP growth and follows a first order Markov Chain. $S = \{s_1, s_2, \ldots, s_t, \ldots\}$ is a discrete set of states, where $t$ stands for time. The value of $s_t$ is defined as $q_i$, which represents certain state of

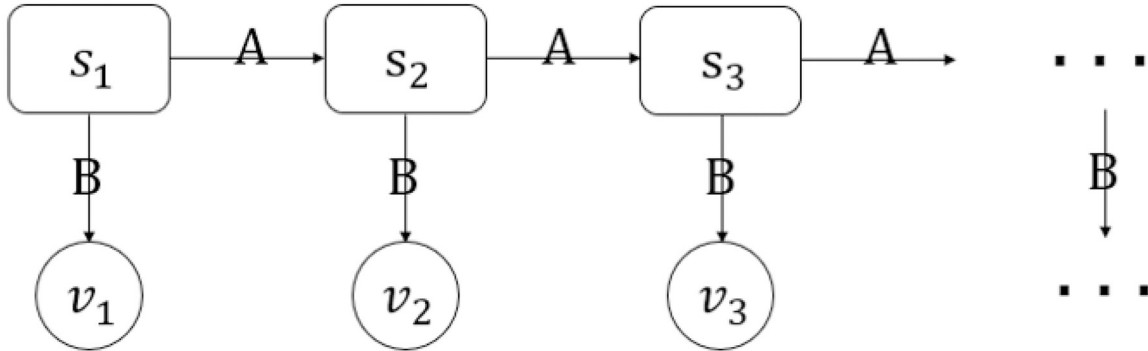

**Fig 1. The structure of the Hidden Markov Model (HMM) that we build as the benchmark.**

GDP growth. The state of GDP growth of next moment $s_{t+1}$ is independent of $s_t$.

$$P(s_{t+1}|s_1, s_2, \ldots, s_t) = P(s_{t+1}|s_t) \tag{1}$$

We set $v$ as an observed variable named CPI fluctuation state, which represents the state of CPI growth rate and forms the observed symbol sequence $V$. $v_t$ represents the CPI fluctuation state observed at time t, which is only related to the corresponding GDP fluctuation state. The value of $v_t$ is defined as $u_k$.

$$P(v_t|v_1, v_2, \ldots, v_{t-1}, s_1, s_2, \ldots, s_t) = P(v_t|s_t) \tag{2}$$

The state transition matrix $A$ is composed of transition probability $a_{ij}$, which represents the prior probability of $s_{t+1}$ depends on $s_t$. $a_{ij}$ is the probability of $s_t$ taking the value of $q_j$ at time $t$ and transferring to $s_{t+1}$ with the value of $q_j$ at time $t + 1$. The matrix $A$ is the transition matrix that represents the probabilities of state transfer composed of $a_{ij}$ calculated based on the observed values. At any time, the GDP fluctuation state $s_{t+1}$, only depends on $s_t$.

$$A = [a_{ij}]_{n*n} \tag{3}$$

$$a_{ij} = P(s_{t+1} = q_i|s_t = q_j) \quad i = 1, 2, \ldots, n; j = 1, 2, \ldots, n \tag{4}$$

In this model, the emission probability matrix $B$ is composed of $b_j(k)$, where $b_j(k)$ is the probability that the CPI fluctuation state $v_t = u_k$ can be observed, provided $s_t = q_j$ when the model is in one of the hidden GDP fluctuation states. The observed CPI fluctuation states are set in Set up the states of CPI fluctuation, where $b_j(k)$ is calculated based on the observable CPI fluctuation states according to their corresponding GDP fluctuation states.

$$B = [b_j(k)]_{n*n} \tag{5}$$

$$b_j(k) = P(v_t = u_k|s_t = q_j) \quad k = 1, 2, \ldots, m; j = 1, 2, \ldots, n \tag{6}$$

The initial state probability vector $\Pi$ is composed of $\pi_i$, where $\pi_i$ is the probability that the GDP fluctuation state $s_1$ will start in state $q_i$ among the GDP fluctuation states.

$$\Pi = [\pi] \tag{7}$$

$$\pi_i = P(s_1 = q_i) \tag{8}$$

To sum up, the parameters of the HMM have been defined and a forecast model of GDP fluctuation based on HMM can be set up, where the HMM parameters {$S$, $V$, $\Pi$, $A$, $B$} are determined based on the historical fluctuation data of GDP and CPI. We further conduct an empirical analysis using data from China to prove the validity of our model. The modeling steps are as follows:

First of all, we define the states of GDP fluctuation based on previous studies [16, 17] in Set up the states of GDP fluctuation, and determine the states of CPI fluctuation according to the threshold model we build in Set up the states of CPI fluctuation, where the CPI fluctuation states may vary with the input variable. We analyze the fluctuation states of these two variables within the sample period and count the numbers of different kinds of states.

Secondly, according to the historical data of GDP growth rate, we make statistics on the GDP fluctuation state sequence $S$ within the time range of empirical data. And then the probability of $s_t$ taking $q_j$ can be calculated and recorded as $\pi_i$, which forms the GDP fluctuation state probability vector $\Pi$.

Thirdly, GDP fluctuation can only be in one state at some point and each state has a certain number of turns, so the value of $q_i$ corresponding to $s_t$ is unique. Thus there are a certain number of transmission paths from $s_t$ to $s_{t+1}$, that is, from value $q_i$ to the next value $q_j$. We can calculate the probability $P_{ij}$ of the transition from $s_t = q_i$ to $s_{t+1} = q_j$ in one of the transition directions, which forms the state transition matrix $A$.

The economic activity is a complex stochastic process, where $s_t$ is invisible at any time. According to the historical data, we can count all the possible $v_t$ corresponding to $s_t$, where $v_t$ forms the observed symbol sequence $V$, which is only related to $S$. Then the emission probability matrix $B$ can be calculated from $S$ to $V$.

Finally, based on the five parameters $\{S, V, \Pi, A, B\}$ calculated, we build the HMM and set out on the empirical work of GDP fluctuation forecast, where the GDP fluctuation state at the next time step can be predicted using Viterbi algorithm as described in the S1 Appendix. The prediction of GDP fluctuation states moves on with the updated data from a rolling time window.

The HMM structure introduced above is an essential component of the models employed in GMM-HMM system and LSTM-HMM structure when predicting GDP fluctuation states. And there are discussions about the input of monthly CPI or quarterly CPI, the lengths of time window and the application of different methods to classifying CPI fluctuation states inThe comparison of forecasting models.

## GMM-HMM system

We build a GMM-HMM system and decompose it into two tasks, including a GMM classifier based on Gaussian Mixture Model that transforms the observed CPI series into a categorical sequence and an HMM model that decode the relationship between CPI and GDP. The major difference of this model with benchmark HMM is the dynamic classification of CPI fluctuation states based on GMM within different time windows, replacing the fixed criteria set for benchmark HMM in Set up the states of the system. In this system, we assume that CPI fluctuation states are following a Gaussian distribution which can be estimated using Expectation-maximization algorithm (EM algorithm), where we set a threshold of 0.001 that determines the stop of training by EM algorithm and the maximum iterations of 1000 times. The Fig 2 below shows the decision process of the GMM-HMM system.

As is shown in Fig 2, we define the input sequence of our forecasting model as $X$ and transform it into sequence $V$ with discrete values using CPI classifier, where the $X$ varies with the CPI series we employ and the $V$ represents the observable CPI fluctuation states of the HMM structure. We firstly obtain the CPI fluctuation states $V$ through the CPI classifier using GMM and then we decode the HMM structure using Viterbi algorithm based on sequence $V$ and $S$,

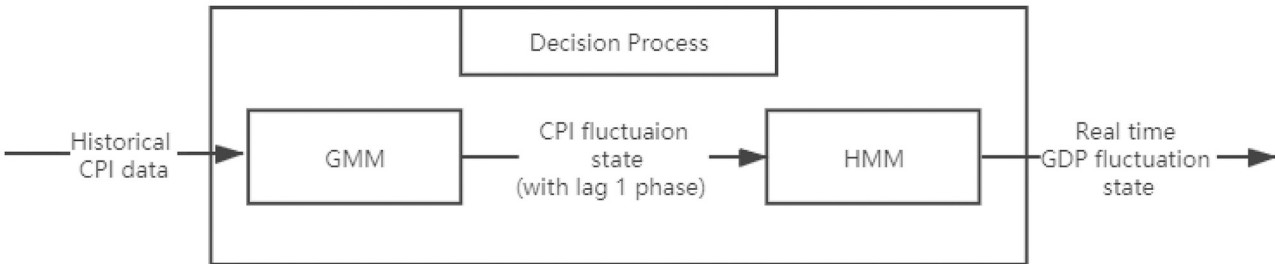

**Fig 2. The decision process of the Gaussian Mixture Model-Hidden Markov Model (GMM-HMM) that introduces Gaussian Mixture Model (GMM) to transform CPI series into a categorical sequence.**

where the transition matrix $A$ and emission probability matrix $B$ can be calculated and applied to the forecast of unobserved GDP fluctuation state at the next time step. The process of decoding the HMM structure is the same with that of The benchmark model: HMM.

The prediction accuracy of the GMM-HMM structure varies with the input and the lengths of time windows. When we predict the GDP fluctuation states using the GMM-HMM system with quarterly CPI series as the input, $X$ involved at each time step is the value of quarterly CPI at lag one period, since it is the latest observable state of a real time GDP fluctuation state. When we predict the GDP fluctuation states using the GMM-HMM system with monthly CPI series as the input, $X$ involved at each time step is a vector of monthly CPI reported in the first two months of a season, since it is the latest observable information of a real time GDP fluctuation state at this time step. The predictions of observable CPI fluctuation state change with the selection of input, since the quarterly CPI at lag one phase represents the economic conditions in the last season, while the monthly CPI reflects the latest information about the first two months of the same season. The lengths of time windows also affect the prediction accuracy, since the number of observations used in training parameters for the GMM-HMM system varies with time windows. The discussions about the input of GMM-HMM system and the selection of time windows are in The comparison of forecasting models.

## LSTM-HMM structure

We build an LSTM-HMM system and decompose it into two tasks, including a forecast model of CPI fluctuation states using Long Short Recurrent Neutral Network (LSTM) based on historical CPI and a forecast model of GDP fluctuation state using HMM. The major difference of this system with the benchmark HMM is that we select the predicted real time CPI fluctuation state by LSTM to be the observable state when we forecast the real time GDP fluctuation state, instead of the observed CPI fluctuation states with time lag. In this system, we train the LSTM using CPI data observed in the past two years and obtain a sequence of discrete values of CPI fluctuation states, where we set a learning rate of 0.001 that determines the stop of iterations and the maximum iterations of 1000 times. The whole decision process of LSTM-HMM structure is showed in Fig 3, with an LSTM structure shown in Fig 4 and a forecast model already shown in Fig 1.

As is shown in Fig 3, we predict the GDP fluctuation state each time the time window moves forward, where we firstly predict the CPI fluctuation state at the same period of GDP that we need to forecast by applying LSTM to the historical CPI series, and then we take the output of LSTM as the updated observable state of the HMM structure to forecast the real time GDP fluctuation state. The details about LSTM involved in this system is shown in Fig 4.

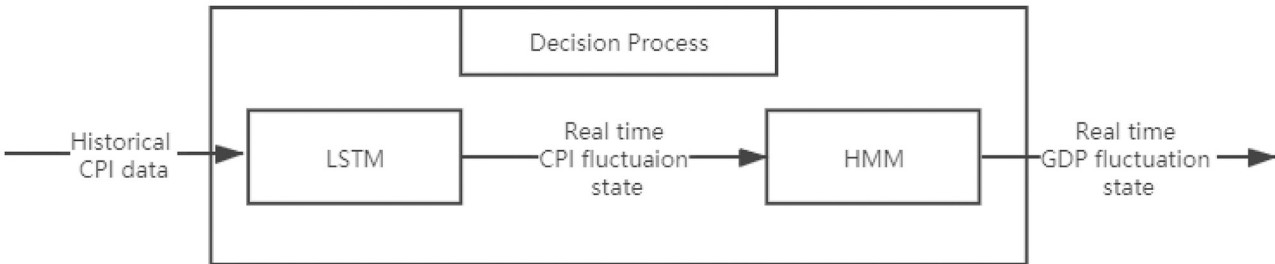

**Fig 3. The whole decision process of Long Short Term Memory Recurrent Neural Network and Hidden Markov Model (LSTM-HMM) with quarterly univariate-input that introduces LSTM to forecast CPI fluctuation states.**

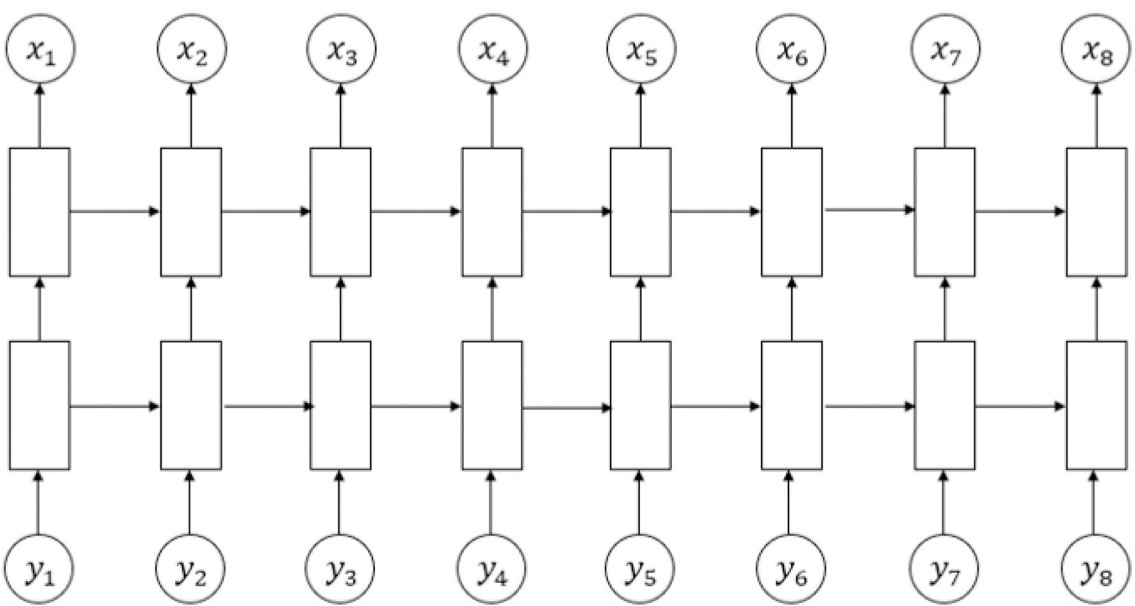

**Fig 4. The structure of the LSTM we build to predict CPI fluctuation states using historical CPI series.**

As we can see from Fig 4, the LSTM neural network consists of three parts: an input neural network layer, an LSTM neural network layer and an output neural network layer, where the input $y_t$ represents historical CPI in the past two years and the output $x_t$ is the real time CPI fluctuation state predicted accordingly. The network of LSTM has 200 hidden units and a learning rate of 0.001, and the maximum number of iterations for each round of prediction is 1000 times. The LSTM function updates the cell and hidden states under the LSTM structure using the hyperbolic tangent function (tanh) as the state activation function and the sigmoid function as the gate activation function. The real time CPI fluctuation state, together with the historical CPI fluctuation states are the input of HMM, and the real time CPI fluctuation state is later applied to the prediction of real time GDP fluctuation states. we predict the real time GDP fluctuation state using real time CPI fluctuation state predicted by LSTM, matrix $A$ and $B$ using Viterbi algorithm. The calculation of the HMM structure is shown in The benchmark model: HMM.

The prediction accuracy of the LSTM-HMM structure also varies with the input and the lengths of time windows. We can predict the real time CPI fluctuation states by LSTM either using quarterly CPI or monthly CPI, each involving 8 observations of quarterly CPI or 24 observations of monthly CPI. The introduction of LSTM ensures that more information about CPI can be used in this system, with an effect on the prediction of observable CPI fluctuation state of HMM structure. The lengths of time windows also affect the prediction accuracy with different numbers of observations used in training parameters. The discussions about the input of LSTM-HMM system and the selection of time windows are in The comparison of forecasting models.

## Empirical analysis

In this section, we describe the variables involved in the empirical analysis in Description of data. We do a cointegration test and a causality relationship test between them in

Cointegration test and Causality relationship test, and build a threshold model between CPI growth rate and GDP growth rate in Threshold model, where the threshold is referred as an indicator of different CPI fluctuation states in certain systems. The detailed criteria of setting states necessary for the forecast systems is in Set up the states of the system, where the criteria of CPI fluctuation states varies with model input and the data processing of forecast systems. In The result of the benchmark HMM, we decode the benchmark HMM with quarterly input and in The comparison of forecasting models, we compare the prediction results of six models with different input and data processing within different time windows, including the benchmark HMM with quarterly input, HMM with monthly input, GMM-HMM with quarterly input, GMM-HMM with monthly input, LSTM-HMM with quarterly input and LSTM-HMM with monthly input. Their prediction results vary with input and time windows of 4-year, 6-year, 8-year and 10-year.

## Description of data

We select CPI as the observed state of HMM to predict GDP fluctuation, where the dynamic relationship between inflation and economic growth is the basis for GDP forecast in this paper. Based on the previous research [8], we learn the synchronization between business and inflation cycles across Chinese provinces. We then further analyze the co-integration between GDP and CPI of China to prove the rationality of variable selection and the research results will serve as a reference for determining criteria of CPI fluctuation state. Our empirical analysis is based on the quarterly data of GDP year-on-year growth rate and the CPI year-on-year growth rate from China between 2000 and 2019, where we use the 80 quarterly observations within twenty years. The data source is China's National Bureau of Statistics (https://data.stats.gov.cn/), a third party website that provides an accessible search engine for National Data of China.

We firstly describe the trend of annual data of CPI and GDP to have a general understanding about their similarity of growth rate. The trend of annual GDP growth rate and CPI growth rate is shown in Fig 5, where the dotted line represents the annual growth rate of CPI and the solid line represents the annual growth rate of GDP. We find that both of them rise on the whole from 2000 to 2008 and then decline from 2008 to 2019. It is obvious that the overall trend of GDP and CPI is similar. However, the local trend of CPI growth rate and the trend of GDP growth rate show no correlation. For example, from 2004 to 2008, GDP rises, but CPI firstly falls and then rises. Therefore, we cannot obtain GDP from CPI only at a certain point in time.

## Cointegration test and causality relationship test

In order to investigate the existence of a stable long-run equilibrium relationship between economic growth and inflation in China, the relationship between GDP and CPI is investigated using the method of linear regression and ADF unit root test based on the data from 2000 to 2019. The linear regression we build is as follows, where we define the quarterly year-on-year growth rate of GDP as $y_t$ and the quarterly data of year-on-year CPI growth rate as $x_t$. The error of regression model is defined as $e_t$, which later serves as the input of ADF unit test. The intercept of our model is defined as $\beta_0$ and the coefficient is defined as $\beta_1$.

$$y_t = \beta_0 + \beta_1 x_t + e_t \tag{9}$$

The estimates of the linear regression model is shown in Table 1, which includes the estimated $\beta_0$ and $\beta_1$. S.E. stands for the abbreviation of standard error and provides information

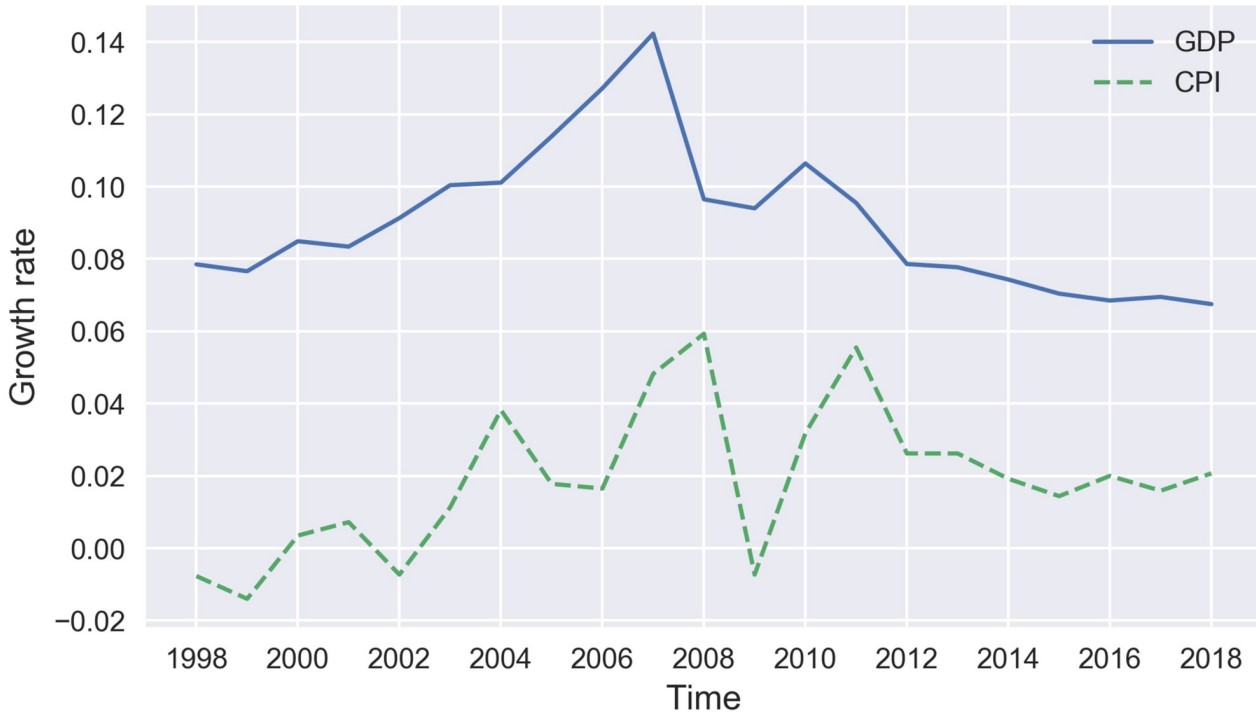

**Fig 5. The trend of GDP year-on-year growth rate and CPI year-on-year growth rate.**

about the accuracy of the estimates. D.F. represents degree of freedom and the P-value shown in the last row of Table 1 shows whether the overall model is significant.

As we can see from Table 1, the coefficient $\beta_0$ is estimated to be 72.49 and $\beta_1$ is estimated to be 0.27, which reach the significance level of 5% and show a positive correlation between $y_t$ and $x_t$, where for every unit change of CPI growth rate, the average change of CPI growth rate is 0.27 units. The degree of freedom represented as D.F. in Table 1 is 79 with 80 samples and one independent variable. Residual S.E. is 1.90. The model has a good goodness of fit with high overall significance since its P-value is only less than 0.01. We then test the stability of the residuals $e_t$ of the linear model to avoid spurious regression using unit root test, where the P-value with lag order of 4 is below the 5% significance level. The result of unit root test shows the stability of the time series $e_t$ involved in our analysis, and we then conduct a cointegration

**Table 1. The linear regression analysis of quarterly GDP growth rate and quarterly year-on-year CPI growth rate.**

| Variables | Coefficients | S.E. | T-value | P-value |
|---|---|---|---|---|
| $\beta_0$ | 72.49 | 10.85 | 6.68 | *** |
| $\beta_1$ | 0.27 | 0.10 | 2.74 | ** |
| | Residual S.E.: 1.90 | D.F.: 79 | P-Value: *** | |

Standard errors are in parentheses.

Significance:

*** $p < 0.01$;

** $p < 0.05$;

* $p < 0.1$.

**Table 2. The result of Granger causality test of quarterly year-on-year GDP growth rate and quarterly year-on-year CPI growth rate.**

| $H_0$ | Lag period | Sample size | F-value | P-value |
|---|---|---|---|---|
| $y_t$ is not the cause of $x_t$ | 5 | 80 | 2.58 | 0.04* |
| $x_t$ is not the cause of $y_t$ | 2 | 80 | 9.79 | *** |

Standard errors are in parentheses.

Significance:

$***p < 0.01$;

$**p < 0.05$;

$*p < 0.1$.

test using method proposed by [18] for the presence of cointegration in the time series of CPI growth rate and GDP growth rate. The result shows that under the significance level of 10 percent, we can reject the hypothesis of no cointegration relationship since the statistical value is 29.83, higher than the critical value of 27.43 at right tail, which indicates the causality between $y_t$ and $x_t$.

We conduct a test of Granger predictive causality with $x_t$ and $y_t$ to obtain the interaction relationship between CPI and GDP in the same period. Considering correlation between two variables that indicates comovement, Granger causality relates to the idea of incremental predictive power of one time series for forecasting another time series, which is a statistically testable criterion based on the ideas of precedence and predictive power [19, 20]. We firstly define the original hypothesis as $H_0$ and the alternative hypothesis as $H_1$. Then we make two hypothesis tests on the cause and effect of $y_t$ on $x_t$ and $x_t$ on $y_t$ respectively, where the optimal lag period is confirmed based on the AIC values of VAR tests. The result is shown in Table 2.

As we can see from Table 2, we can reject either $H_0$, which shows evidence for Granger causality in both directions.

## Threshold model

For further analysis of the relationship between CPI and GDP, we then introduce an instrumental variable which is a reduced form equation of $x_t$. We define the instrumental variable as $h_t$, and then select CPI data with one phase lag due to its relevance with $x_t$ and irrelevance with $y_t$. To verify the rationality of the selected $h_t$, we make a correlation analysis between $x_t$ and $h_t$, $y_t$ and $h_t$. The result is shown in Table 3.

As we can see from Table 3, the correlation of $x_t$ and $y_t$ is 0.88 with a P-value less than 1%. However, $h_t$ is not correlated with $y_t$ since the P-value of correlation analysis is 0.42, more than the significance level of 5%. Therefore, we determine to build a reduced form equation of $x_t$ using the instrumental variable $h_t$. Based on the literatures of threshold effect between economic growth and inflation, we build a regression kink model with endogenous variables but an exogenous threshold variable, where we develop a two-stage least squares estimator of the threshold parameter and a generalized method of moments estimator of the slope parameter [13]. The structural equation is

$$y_t = \theta_1^{'} z_t I\{h_t \leq \gamma\} + \theta_2^{'} z_t I\{h_t > \gamma\} + e_t \qquad (10)$$

The observed sample is $\{y_t, z_t, x_t\}$, where $y_t$ stands for GDP, $z_t$ is a vector including $x_t$ and constant $\alpha$, where $x_t$ is a vector of CPI in the same period as GDP. The threshold variable $h_t = h(x_t)$ is an element or function of the vector $x_t$, where we set $h_t$ as CPI data with one phase lag.

**Table 3. The result of correlation analysis of quarterly CPI with one phase lag and quarterly GDP.**

| Indexes | $x_t$ v.s.$h_t$ | $y_t$ v.s.$h_t$ |
|---|---|---|
| T-value | 15.88 | 0.80 |
| D.F. | 79 | 79 |
| Correlation | 0.88 | 0.09 |
| P-value | *** | 0.43 |

Standard errors are in parentheses.

Significance:

***$p < 0.01$;

**$p < 0.05$;

*$p < 0.1$.

The model allows the slope parameters differ depending on the value of $h_t$. We define regime1 for samples falling into the category less than the estimated threshold and regime 2 for samples larger than the threshold. $\theta_1^{'}$ is a vector of parameters for samples in regime 1 and $\theta_2^{'}$ is for samples in regime 2. We estimate the parameters sequentially using the method employed in previous studies [9]. Firstly, we estimate the threshold of our model using the method of least square errors and calculate the confidence intervals of this parameter, where the threshold estimate has the same distribution as for the regression case with a different scale [12].

To set up the confidence interval $\gamma$, a standard approach is to use the likelihood ratio under the auxiliary assumption that $e_t$ is iid $N(0, \delta^2)$, where the uncorrected confidence interval is built using asymptotic critical values based on the likelihood ratio statistic.

$$LR_n(\gamma) = n \frac{S_n(\gamma) - S_n(\hat{\gamma})}{S_n(\hat{\gamma})} \qquad (11)$$

However, if the homoskedasticity condition does not hold, a heteroskedasticity-robust asymptotic 90%-level confidence region for $\gamma$ based on a scaled likelihood ratio statistic is introduced as Confidence Interval—Heteroskedasticity Corrected 1 (C.I.—Het Corrected 1). Another theorem of corrected confidence interval with heteroskedasticity is introduced in the case of Gaussian errors, where the likelihood ratio test is asymptotically conservative and the amended confidence interval is defined as Confidence Interval—Heteroskedasticity Corrected 2 (C.I.—Het Corrected 2) [13]. The confidence interval with uncorrected heteroskedasticity (C.I.—Uncorrected) is shown below. All the related estimates about threshold are shown in Fig 6 and Table 4, where C.I. is an abbreviation of confidence interval.

As we can see from Fig 6, the threshold estimates of this model is the value that minimizes this graph, which occurs at $\hat{\gamma} = 101.37$. The 90% confidence intervals under different scales are also plotted(the dotted line) respectively, so we can read off the asymptotic 90% confidence set [101.33,102.67] from the graph from where test statistics cross the dotted line. where $\hat{\gamma}$ is significant when taking the value of 101.37. As is shown in Table 4, the confidence intervals of different scales are the same with the threshold estimate of 101.37. These results show that there is reasonable evidence for a two-regime specification.

Given the estimate $\hat{\gamma}$ of the threshold $\gamma$, the sample can be split into two sub-samples based on the indicators $I\{h_t \leq \gamma\}$ and $I\{h_t > \gamma\}$, where samples with the threshold variable of CPI growth rate with one phase lag less or greater than $\gamma$ are each divided into two groups. The slope parameters $\theta_1$ and $\theta_2$ can be estimated by 2SLS separately on each subsample. In Table 5, $\theta_1$ and $\theta_2$ stands for the slope coefficients, and $\alpha_1$ and $\alpha_2$ stands for the estimated intercepts. $n_1$

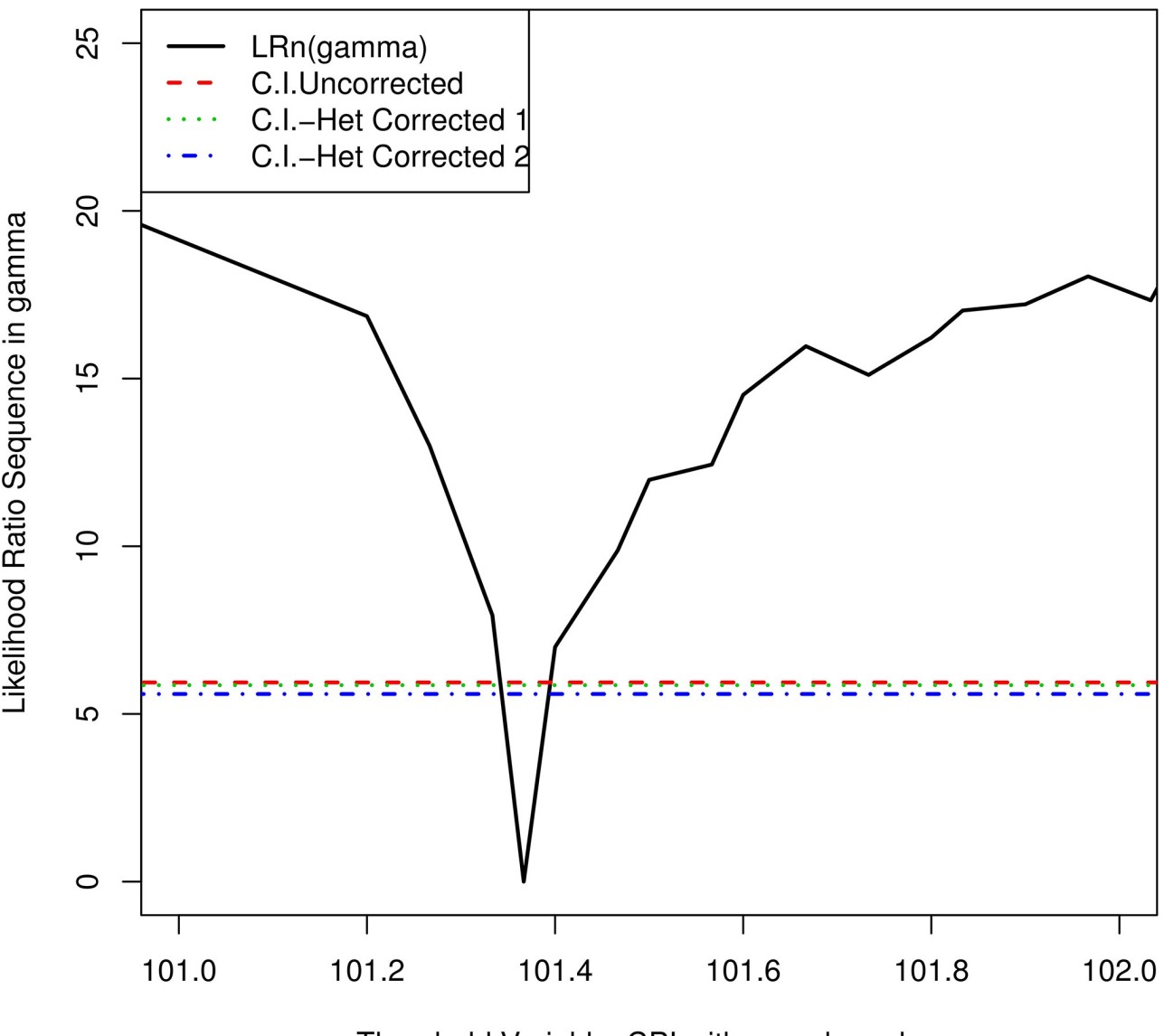

**Fig 6. The trend of GDP year-on-year growth rate and CPI year-on-year growth rate.**

**Table 4. The threshold estimates from two-stage least squares estimation for the threshold model.**

| Threshold estimate $\hat{\gamma}$ | 101.37 |
|---|---|
| C.I.—Uncorrected | [101.33,102.67] |
| C.I.—Het Corrected 1 | [101.33,102.67] |
| C.I.—Het Corrected 2 | [101.33,102.67] |

**Table 5. Least squares estimates of 2-regime threshold model that we build with CPI and GDP data.**

| Coefficient | Estimate | S.E. | C.I. |
|---|---|---|---|
| $\alpha_1$ | 77.95 | 45.20 | [-10.65,166.55] |
| $\theta_1$ | 0.32 | 0.45 | [-0.57,1.20] |
| $\alpha_2$ | 45.30 *** | 12.93 | [19.96,70.65] |
| $\theta_2$ | 0.62 *** | 0.12 | [0.37,0.86] |

Standard errors are in parentheses.

Significance:

***$p < 0.01$;

**$p < 0.05$;

*$p < 0.1$.

and $n_2$ each represents the number of samples falling into each subsample. The dynamic relationship between $y_t$ and $x_t$ is shown in Table 5.

As we can see from Table 5, the model allows the slope parameters to differ depending on the value of $h_t$. The slope coefficient $\theta_1$ is 0.32 given $h_t \leq 101.37$, while the slope coefficient $\theta_2$ is 0.62 given $h_t \geq 101.37$. There are 25 observations of samples in regime 1 and 55 observations of samples in regime 2, each accounting for one quarter and three quarters of our sample data.

The findings above further confirm that GDP growth and inflation tend to move in a similar trend and reveal the threshold effect between GDP and CPI, which serves as a reference for the determination of CPI fluctuation state. Therefore in this paper, we select CPI observed to predict GDP fluctuation state.

## Set up the states of the system

We build six models based on the forecast systems introduced in The econometric model and estimation method we employ, where we firstly need to define a criteria to describe the states of GDP fluctuation and CPI fluctuation. The criteria of GDP fluctuation states is applicable to all the models involved in this paper, while the criteria of CPI fluctuation states varies with models and their input.

**Set up the states of GDP fluctuation.** Firstly, we set $s$ as a "three states" unobserved variable following a first order Markov Chain, and then we make statistics on $s_t$ within the time range of empirical data. Referring to the study on the trend of China's macroeconomic indicators [16], we use a threshold autoregression model to estimate the turning points of the GDP growth rate curve between 2000 and 2019. We represent the GDP growth rate at time $t$ with $y_t$ and set the corresponding threshold variable as $z_t = y_{t-d}$ with a delay order $d(d \leq p)$. The TAR model is defined as

$$y_t = (\beta_{10} + \beta_{11}y_{t-1} + \ldots + \beta_{1p}y_{t-p} + \sigma_1\varepsilon_t)I_{(z_t \leq c_1)}$$
$$+(\beta_{20} + \beta_{21}y_{t-1} + \ldots + \beta_{2p}y_{t-p} + \sigma_2\varepsilon_t)I_{(c_1 < z_t \leq c_2)}$$
$$+(\beta_{30} + \beta_{31}y_{t-1} + \ldots + \beta_{3p}y_{t-p} + \sigma_1\varepsilon_t)I_{(z_t > c_3)}$$

where the $c_1$ and $c_2$ each represents the turning points of GDP growth rate. We set $p$ as 8 and estimate the parameters including the delay order $d$, the turning points $c_1$ and $c_2$, as well as the coefficients of the variables. The results are shown in Table 6.

As we can see from Table 6, the delay order $p$ is estimated to be 8. The values of the threshold are estimated as 107.5 and 108.9. The sum of squared errors of the estimated TAR is 40.94.

**Table 6. The results of the threshold auto-regression model of the GDP growth rate.**

| Variable | Regime 1 ($z_t \leq 107.50$) | | Regime 2 ($107.50 < z_t \leq 108.90$) | | Regime 3 ($z_t > 108.90$) | |
|---|---|---|---|---|---|---|
| | Estimate | S.E. | Estimate | S.E. | Estimate | S.E. |
| Constant | -63.93 *** | 20.50 | -116.72 *** | 34.59 | 5.43 | 13.01 |
| $y_{t-1}$ | 0.73 *** | 0.17 | -0.20 | 0.21 | 1.21*** | 0.14 |
| $y_{t-2}$ | 0.61 *** | 0.12 | 0.48 *** | 0.13 | -0.43 | 0.24 |
| $y_{t-3}$ | -0.19 | 0.16 | 0.35** | 0.12 | 0.08 | 0.33 |
| $y_{t-4}$ | -1.29 *** | 0.17 | -0.02 | 0.20 | -0.08 | 0.30 |
| $y_{t-5}$ | 1.29 *** | 0.26 | 0.39 | 0.27 | 0.05 | 0.24 |
| $y_{t-6}$ | 0.58 | 0.34 | 0.12 | 0.21 | -0.05 | 0.25 |
| $y_{t-7}$ | -0.93 *** | 0.17 | -0.33*** | 0.11 | -0.27 | 0.22 |
| $y_{t-8}$ | 0.80 *** | 0.19 | 1.29*** | 0.38 | 0.42** | 0.15 |
| | Delay Order $d$ | 8 | SSE | 40.94 | F value | 36.70 |

Standard errors are in parentheses.

Significance:

*** $p < 0.01$;

** $p < 0.05$;

* $p < 0.1$.

Referring to [21], the test for the null hypothesis of a one-regime auto-regression (AR) against the alternative hypothesis of a three-regime TAR are conducted. The F value is calculated as 36.69, higher than the 99% significance level, indicating the rejection of the null hypothesis of AR.

We divide the state of economic fluctuation into low-speed growth stage, appropriate-speed growth stage and high-speed growth stage with the turning points of the GDP growth rate curve as the dividing points, which is 7.50% and 8.90% respectively. We then mark the above three States as 0, 1 and 2. As is shown in Table 7.

Based on the above standards, GDP fluctuation state sequence $S$ is constructed according to the quarterly data of year-on-year growth rate of GDP from 2000 to 2019.

**Set up the states of CPI fluctuation.** In this paper, we introduce six models with different input, including benchmark HMM with quarterly input, HMM with monthly input, GMM-HMM with quarterly input, GMM-HMM with monthly input, LSTM-HMM with quarterly input and LSTM-HMM with monthly input, and these forecast systems vary in the input and the criteria of CPI fluctuation states.

The criteria set in Table 8 based on our research results of threshold model in Threshold model is adopted by the benchmark HMM with quarterly univariate-input, LSTM-HMM with quarterly input and LSTM with monthly input, where the CPI fluctuation states obtained in this way is taken as the input of benchmark HMM and the labels necessary for the training of LSTM classification problem in the LSTM-HMM system. This is a general criteria for the CPI

**Table 7. GDP fluctuation states that differ in three types.**

| Sequence value | GDP fluctuation state |
|---|---|
| 0 | Low-speed growth stage |
| 1 | Appropriate-speed growth stage |
| 2 | High-speed growth stage |

**Table 8. A general criteria for CPI fluctuation states with two different types.**

| Sequence value | Confirmation standard | CPI fluctuation state |
|---|---|---|
| 0 | $x \leq 101.37$ | Moderate inflation |
| 1 | $x > 101.37$ | Intensified inflation |

**Table 9. Criteria of CPI fluctuation states for HMM with monthly input that vary in four types.**

| Sequence value | Confirmation standard | | CPI fluctuation state |
|---|---|---|---|
| 0 | $x_{t1} \leq 101.37$ | $x_{t2} \leq 101.3667$ | Moderate-Moderate inflation |
| 1 | $x_{t1} \leq 101.37$ | $x_{t2} > 101.3667$ | Moderate-Intensified inflation |
| 2 | $x_{t1} > 101.37$ | $x_{t2} \leq 101.3667$ | Intensified-Moderate inflation |
| 3 | $x_{t1} > 101.37$ | $x_{t2} > 101.3667$ | Intensified-Intensified inflation |

fluctuation state obtained from recently observed quarterly CPI with lag one period for benchmark HMM, and it is also a criteria for real time CPI fluctuation states applied to the training of LSTM. Therefore, we represent the CPI involved in forecast systems with $x$ in Table 8 without any denote for a general classification. As we can see from Table 8, we divide the CPI fluctuation states $v_t$ into two regimes at 101.3667, which is regime 1 defined as moderate inflation and regime 2 defined as intensified inflation. We mark the above two states as 0 and 1.

As for CPI fluctuation states obtained from monthly data, we define the CPI fluctuation states considering the order and values of the two monthly CPI observed for each GDP fluctuation state, which is shown in Table 9. This is a criteria for the CPI fluctuation state obtained from recently observed monthly CPI in the first two months of a season especially set for HMM with monthly input. Therefore, we represent the historical monthly CPI involved in HMM forecast systems with $x_{t1}$ and $x_{t2}$ in Table 8 and divide the observable CPI fluctuation states into four states referring to the threshold in Threshold model. As we can see from Table 9, we further divide the states of CPI fluctuation obtained from monthly CPI into four stages, which is marked as 0, 1, 2 and 3 with Moderate-Moderate inflation, Moderate-Intensified inflation, Intensified-Moderate inflation and Intensified-Intensified inflation.

As for the CPI fluctuation states employed in GMM-HMM system, we generate the CPI fluctuation state sequence in each round of training using GMM classifier, where we either introduce a sequence of quarterly CPI or a sequence of monthly CPI vector containing monthly CPI observed in the first two months of the current season. The details about the GMM classifier are in GMM-HMM system.

To sum up, we have set the GDP fluctuation states and the fixed criteria for CPI fluctuation states involved in HMM system and LSTM-HMM system, while the CPI fluctuation states in GMM-HMM system are generated using GMM classifier in each round of training within the time window. There are two kinds of states for quarterly CPI series and four states for monthly CPI series.

**The relationship between GDP fluctuation state and CPI fluctuation state.** We firstly analyze the relationship between GDP fluctuation states and CPI fluctuation states obtained from fixed criteria by introducing the emission probability matrix calculated based on the quarterly data set involved in this paper, which shows how GDP influences the movement of

CPI. The emission probability matrix is shown as follows:

$$\begin{pmatrix} 0.14 & 0.86 \\ 0.35 & 0.65 \\ 0.46 & 0.54 \end{pmatrix}$$

where given GDP fluctuation state taking different values, the CPI fluctuation states are more likely to take value 1, which means that the fluctuation of CPI series is not moderate in most cases. The emission probabilities of CPI taking value 1 given different GDP fluctuation states vary, while the emission probability of CPI fluctuation states taking value 1 given GDP fluctuation state of 0 is the highest among other emission probabilities within the matrix.

We then analyze the relationship between CPI fluctuation states and GDP fluctuation states by introducing the emission probability matrix calculated based on the monthly CPI series involved in this paper, which shows how GDP influences the movement of CPI. The emission probability matrix is shown as follows:

$$\begin{pmatrix} 0.04 & 0.11 & 0.14 & 0.71 \\ 0.35 & 0.00 & 0.00 & 0.65 \\ 0.37 & 0.03 & 0.06 & 0.54 \end{pmatrix}$$

where given GDP fluctuation state taking different values, the CPI series are more likely to be in the stage of "Intensified-Intensified inflation". Given GDP fluctuation states taking the value of 1, CPI series are more likely to be in the stage of "Moderate-Moderate inflation" and "Intensified-Intensified inflation" with no samples taking the value of 1 or 2. Given different GDP fluctuation states, the probabilities of CPI fluctuation states taking the value of 1 or 2 is small. which means that the probability of transition of monthly CPI fluctuation states between moderate state and intensified state is lower than the probability of keeping in one state.

## The result of the benchmark HMM

In this subsection, we introduce the HMM with quarterly input within 8-year window and show its predictions of GDP fluctuation states. This model has best performance among the benchmark hidden Markov models with quarterly input within different time windows not only in accuracy but also in consistency, which is discussed and testified with the other models derived from benchmark HMM within different time windows in The comparison of forecasting models.

We decode the HMM structure using observations from 2000 to 2019, where we set a time window of 4 years with 16 observations. Every time we roll the time window, the data involved is updated and a new observation of CPI fluctuation state is available. And then we are able to predict the GDP fluctuation state at next time step. There are 64 predictions of GDP fluctuation states in total, and we compare them with the real values, where the confusion matrix is shown below in Fig 8. Fig 7 below shows the predicted values and the real values.

As we can see from Fig 7, there are 64 predictions in total, ranging from 2004.Q1 to 2019. Q4. There are 41 predictions coinciding with the real values, where there are 17 accurate predictions of GDP fluctuation states with the value of 0, 9 accurate predictions of GDP fluctuation states of value 1 and 12 accurate predictions of GDP fluctuation states of value2, while there are 26 wrong predictions evenly distributed. The predictions after 2012 have a better performance than the predictions before that time and the two curves in Fig 7 fit with each other.

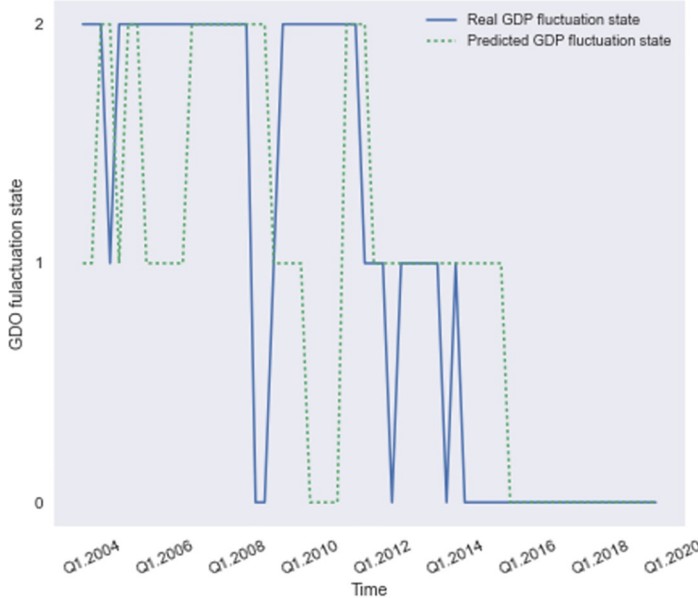

**Fig 7. The predictions of GDP fluctuation states using HMM(q) within 4-year time window.**

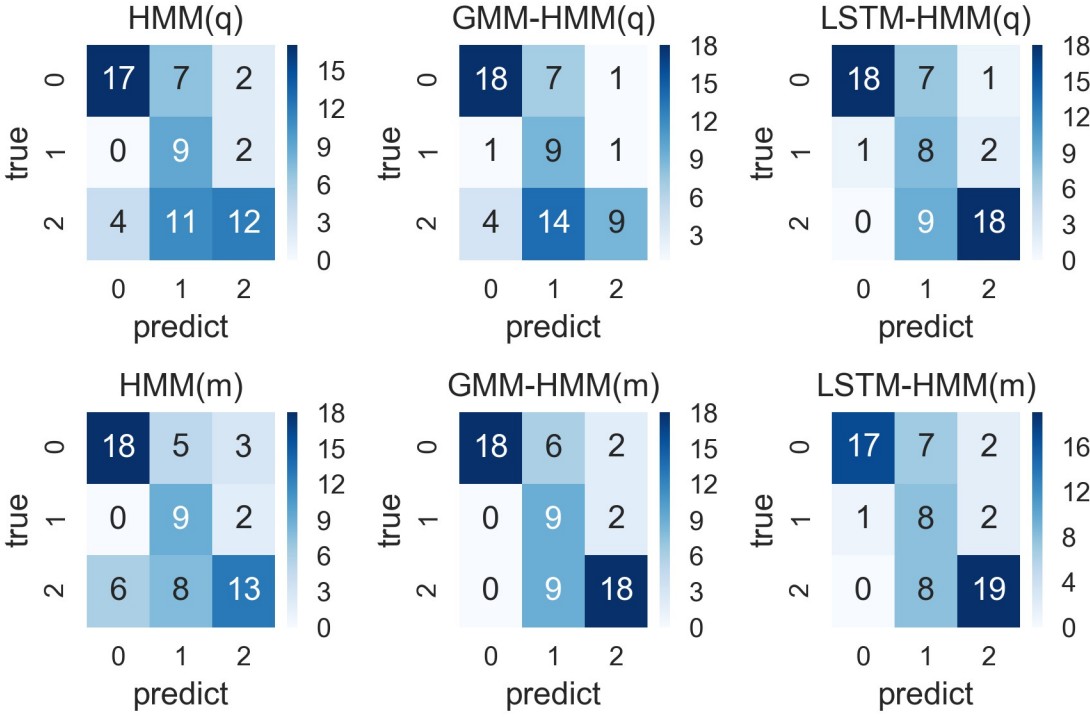

**Fig 8. Confusion matrixes for prediction results from HMM, GMM-HMM and LSTM-HMM with monthly or quarterly input and 4-year time window.**

To better evaluate the prediction capability of the forecasting models, we then train the other models using various methods based on data from different time windows.

## The comparison of forecasting models

We introduce another six models to compare the prediction accuracy of GDP fluctuation, where these models vary in the model inputs and estimating methods. And we apply them to different observations within different time windows, and also compare the parameters about prediction accuracy. The models we compare are HMM with quarterly input, GMM-HMM with quarterly input, LSTM-HMM with quarterly input and HMM with monthly input, GMM-HMM with monthly input and LSTM-HMM with monthly input, each represented as HMM(q), GMM-HMM(q), LSTM-HMM(q), HMM(m), GMM-HMM(m), LSTM-HMM(m) in our empirical analyses within time windows of 4-year (2000–2003), 6-year (2000–2005), 8-year (2000–2007) and 10-year (2000–2009) with 16, 24, 32, 40 observations of quarterly CPI, 32, 48, 64 and 80 observations of monthly CPI, as well as 64, 56, 48 and 40 predictions of quarterly GDP fluctuation states. The analysis about these models are based on historical data within a period of 20 years from 2000 to 2019, where we split the training set and test set according to the rolling time window. The details about observations and the predictions are shown in Table 10.

We run the six models and compare them within each time window respectively. Their prediction results are shown in confusion matrixes in Figs 8–11. We also define a accuracy vector with the parameters of accuracy, kappa and AUC, which are calculated to evaluate the forecast model based on the confusion matrix, and the accuracy vectors involved in the comparison are detailed in Tables 11–14, and summarized in Comparing models within different time windows. The accuracy rate represents the prediction accuracy, and the kappa is used for consistency test and evaluating the accuracy of classification. The range of kappa is from 0 to 1, which can be divided into five groups to represent different levels of consistency: $0.00 \sim 0.20$ for slight consistency, $0.21 \sim 0.40$ for fair consistency, $0.41 \sim 0.60$ for moderate consistency, $0.61 \sim 0.80$ for substantial consistency and $0.81 \sim 1.00$ for almost perfect consistency. AUC (area under curve) is defined as the area enclosed by the coordinate axis under ROC curve, where AUC value is between 0.50 and 1.00. The ROC and AUC involved in this paper are calculated using "micro-average" method from "sklearn" package for python. The closer AUC is to 1.00, the higher the authenticity of the detection method is; when AUC is equal to 0.50, the lower the authenticity and no application value. The model with higher accuracy, higher AUC and higher accuracy will have a better performance in the prediction of classification. We finally compare the prediction capability of all the forecasting models in Comparing models within different time windows according to their ROC in Fig 12 and their evaluating vectors in Table 15.

**Models within 4-year time window.** We run the six models including HMM(q) with quarterly input, GMM-HMM(q) with quarterly input, LSTM-HMM(q) with quarterly input,

**Table 10. The number of observations and predictions of GDP fluctuation states within 4-year, 6-year, 8-year and 10-year time window.**

| Time window | Forecasting period | Observations(Q) | Observations(M) | Predictions |
|---|---|---|---|---|
| 4-year (2000–2003) | 16-year (2004–2019) | 16 | 32 | 64 |
| 6-year (2000–2005) | 14-year (2006–2019) | 24 | 48 | 56 |
| 8-year (2000–2007) | 12-year (2008–2019) | 32 | 64 | 48 |
| 10-year (2000–2009) | 10-year (2010–2019) | 40 | 80 | 40 |

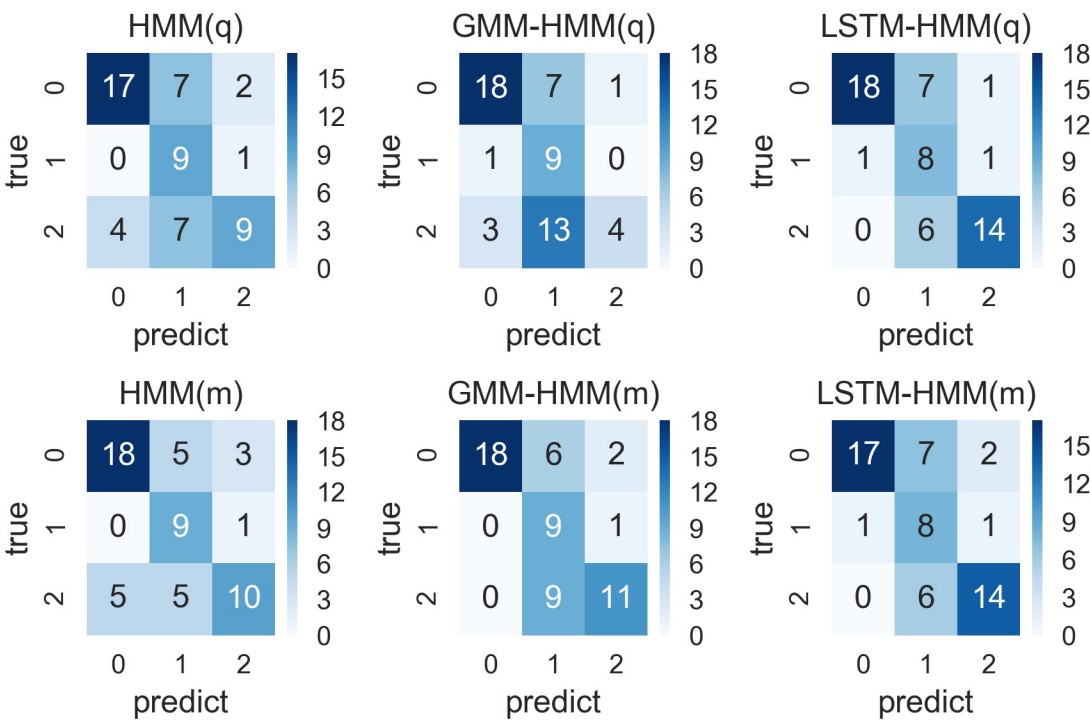

**Fig 9. Confusion matrixes for prediction results from HMM, GMM-HMM and LSTM-HMM with monthly or quarterly input and 6-year time window.**

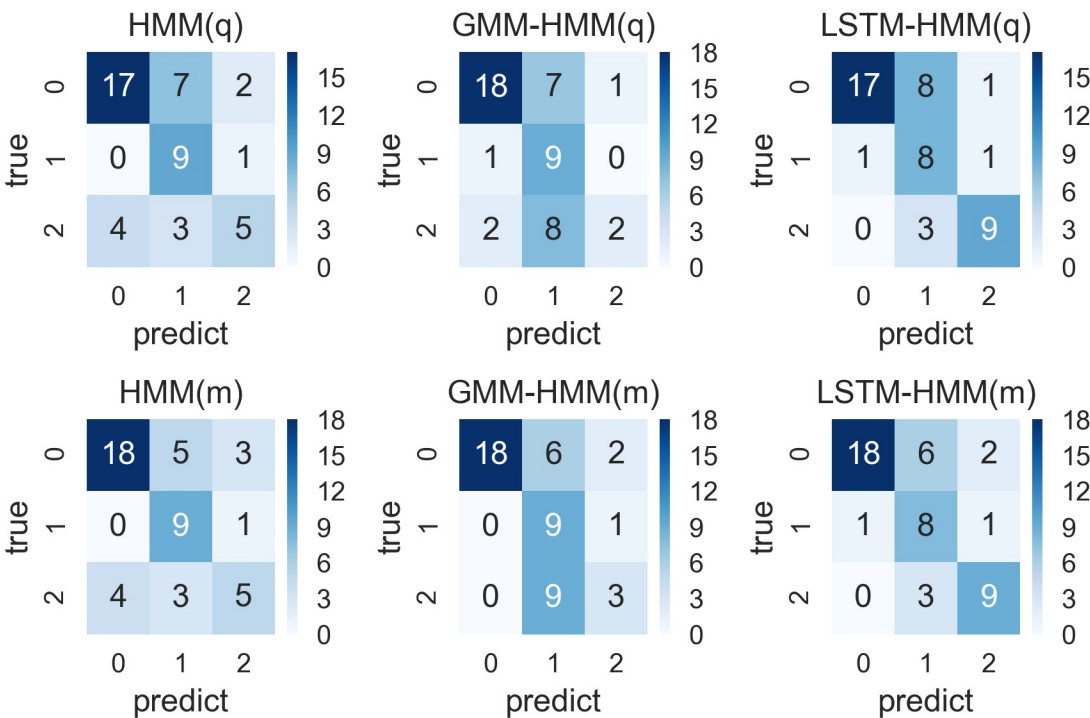

**Fig 10. Confusion matrixes for prediction results from HMM, GMM-HMM and LSTM-HMM with monthly or quarterly input and 8-year time window.**

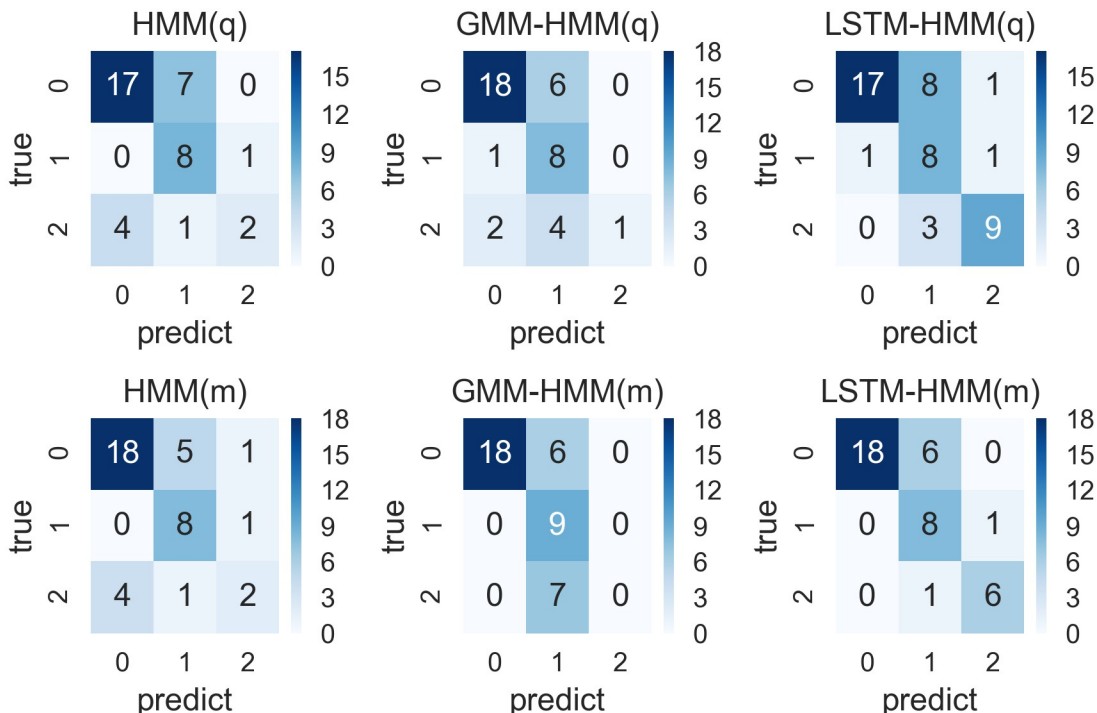

**Fig 11. Confusion matrixes for prediction results from HMM, GMM-HMM and LSTM-HMM with monthly or quarterly input and 10-year time window.**

HMM(m) with monthly input, GMM-HMM(m) with monthly input and LSTM-HMM(m) with monthly input within 4-year time windows and their confusion matrixes calculated based on the predictions and real values are shown in Fig 8.

As we can see from the Fig 8, there are 64 predictions in total, where there are many predictions of GDP fluctuation states with value 0 and 2. All these models have precise predictions of GDP fluctuation states with value 0, while GMM-HMM with monthly input and LSTM-HMM with either monthly or quarterly input also have good performance in the prediction of GDP

**Table 11. Accuracy, kappa and AUC for HMM, GMM-HMM and LSTM-HMM with monthly or quarterly input and 4-year time window.**

|  | HMM(q) | GMM-HMM(q) | LSTM-HMM(q) | HMM(m) | GMM-HMM(m) | LSTM-HMM(m) |
|---|---|---|---|---|---|---|
| Accuracy | 0.59 | 0.56 | 0.69 | 0.63 | 0.70 | 0.69 |
| Kappa | 0.41 | 0.38 | 0.54 | 0.44 | 0.56 | 0.54 |
| AUC | 0.70 | 0.67 | 0.77 | 0.72 | 0.78 | 0.77 |

**Table 12. Accuracy, kappa and AUC for HMM, GMM-HMM and LSTM-HMM with monthly or quarterly input and 6-year time window.**

|  | HMM(q) | GMM-HMM(q) | LSTM-HMM(q) | HMM(m) | GMM-HMM(m) | LSTM-HMM(m) |
|---|---|---|---|---|---|---|
| Accuracy | 0.63 | 0.55 | 0.71 | 0.66 | 0.68 | 0.70 |
| Kappa | 0.45 | 0.36 | 0.58 | 0.49 | 0.53 | 0.55 |
| AUC | 0.72 | 0.67 | 0.79 | 0.75 | 0.76 | 0.77 |

**Table 13. Accuracy, kappa and AUC for HMM, GMM-HMM and LSTM-HMM with monthly or quarterly input and 8-year time window.**

|  | HMM(q) | GMM-HMM(q) | LSTM-HMM(q) | HMM(m) | GMM-HMM(m) | LSTM-HMM(m) |
|---|---|---|---|---|---|---|
| Accuracy | 0.65 | 0.60 | 0.71 | 0.67 | 0.63 | 0.73 |
| Kappa | 0.45 | 0.38 | 0.56 | 0.47 | 0.43 | 0.58 |
| AUC | 0.73 | 0.70 | 0.78 | 0.75 | 0.72 | 0.80 |

**Table 14. Accuracy, kappa and AUC for HMM, GMM-HMM and LSTM-HMM with monthly or quarterly input and 10-year time window.**

|  | HMM(q) | GMM-HMM(q) | LSTM-HMM(q) | HMM(m) | GMM-HMM(m) | LSTM-HMM(m) |
|---|---|---|---|---|---|---|
| Accuracy | 0.68 | 0.68 | 0.80 | 0.70 | 0.68 | 0.80 |
| Kappa | 0.44 | 0.44 | 0.67 | 0.48 | 0.46 | 0.67 |
| AUC | 0.76 | 0.76 | 0.85 | 0.78 | 0.76 | 0.85 |

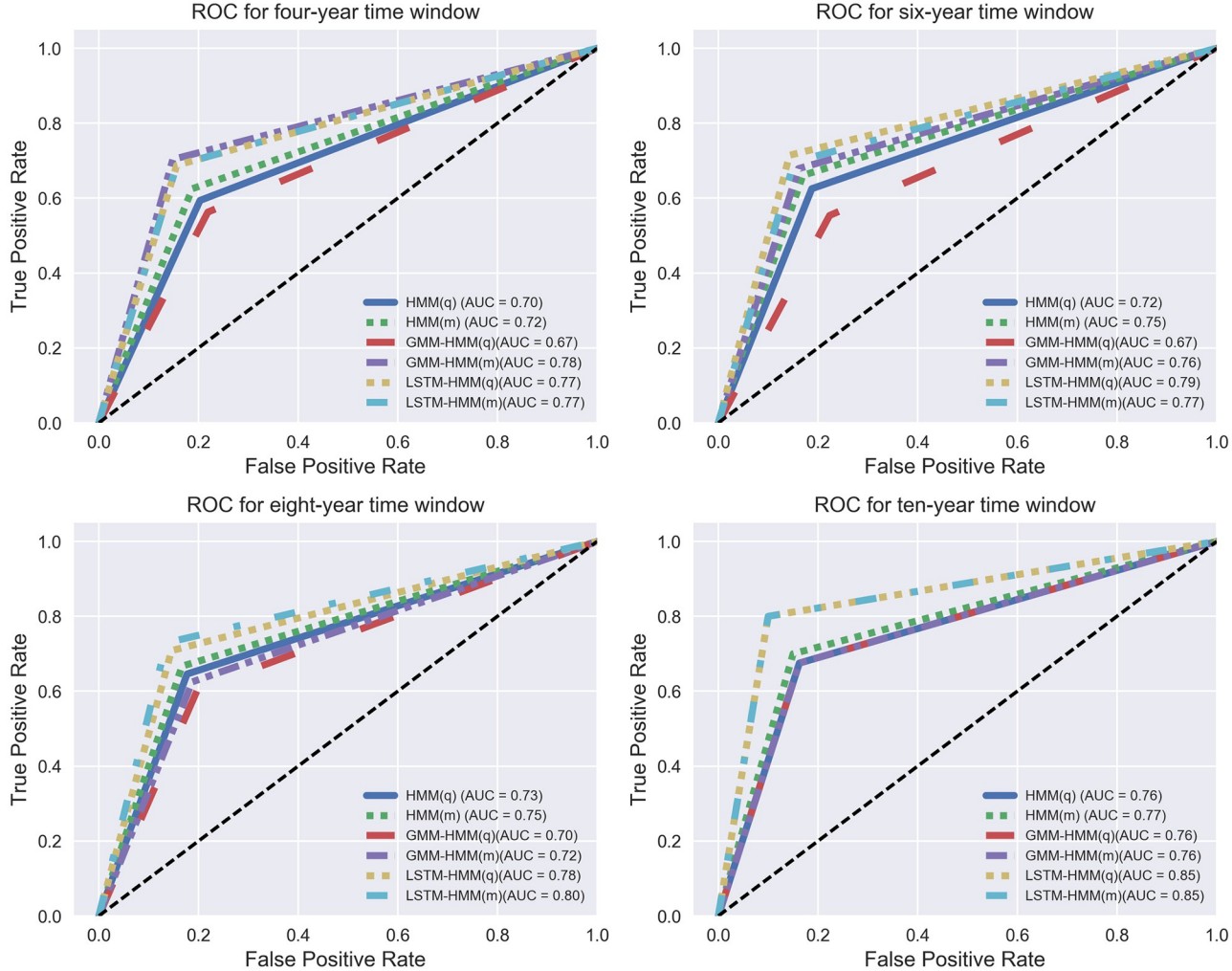

**Fig 12. ROC for HMM, GMM-HMM and LSTM-HMM with monthly or quarterly input within 4-year, 6-year, 8-year and 10-year time windows.**

**Table 15. Comparing HMM, GMM-HMM and LSTM-HMM with monthly or quarterly input and 4-year, 6-year, 8-year or10-year time windows based on Accuracy, Kappa and AUC.**

| T.W. | HMM(q) | GMM-HMM(q) | LSTM-HMM(q) | HMM(m) | GMM-HMM(m) | LSTM-HMM(m) |
|---|---|---|---|---|---|---|
| 4 year | (0.59,0.41,0.70) | (0.56,0.38,0.67) | (0.69,0.54,0.77) | (0.63,0.44,0.72) | (0.70,0.56,0.78) | (0.69,0.54,0.77) |
| 6 year | (0.63,0.45,0.72) | (0.55,0.36,0.67) | (0.71,0.58,0.79) | (0.66,0.49,0.75) | (0.68,0.53,0.76) | (0.70,0.55,0.77) |
| 8 year | (0.65,0.455,0.73) | (0.60,0.38,0.70) | (0.71,0.56,0.78) | (0.67,0.47,0.75) | (0.63,0.43,0.72) | (0.73,0.58,0.80) |
| 10 year | (0.68,0.44,0.76) | (0.68,0.44,0.76) | (0.80,0.67,0.85) | (0.70,0.48,0.78) | (0.68,0.46,0.76) | (0.80,0.67,0.85) |

fluctuation state 1. From the perspective of overall efficiency, we can see that GMM-HMM with monthly input has 45 accurate predictions. This is more than that of other predictions and is slightly better than the performance of LSTM-HMM. Based on the confusion matrixes shown in Fig 8, we calculated the accuracy vector necessary for the evaluation, including accuracy, kappa and AUC, where these details are shown in Table 11. The models with monthly input are denoted with m, while the models with quarterly input are denoted with q.

As is shown in Table 11, the GMM-HMM with monthly input has the highest prediction accuracy, which is 0.70, with the highest kappa of 0.56 and the highest AUC of 0.78, showing a moderate consistency and a good performance in prediction. HMM(m), LSTM-HMM(q) and LSTM-HMM(m) also perform better than random judgement, with a prediction accuracy over 0.60, while HMM(q) and GMM-HMM(q) do not have a satisfactory performance in prediction, with a lower prediction accuracy, kappa and AUC. When the model input transforms from quarterly to monthly, the model performance of the HMM and GMM-HMM can be improved in terms of accuracy, kappa and AUC, while the LSTM-HMM, with either monthly or quarterly input, keeps the same model performance.

To sum up, within the 4-year time window which involves 16 observations for each round of training, the GMM-HMM(m) with quarterly input has the best performance, while the HMM(q) and GMM-HMM(q) with quarterly input may not be applicable to the prediction in this condition.

**Models within 6-year time window.** We run the six models including HMM(q), GMM-HMM(q), LSTM-HMM(q), HMM(m), GMM-HMM(m) and LSTM-HMM(m) within 6-year time windows and their confusion matrixes are shown in Fig 9.

As we can see from the Fig 9, there are 56 predictions in total, and the characteristics of these confusion matrixes are similar with that of models within 4-year time window. All these models predict many GDP fluctuation states with value 0. The GMM-HMM(m), LSTM-HMM(q), LSTM-HMM(m) have relatively better performance in the prediction of GDP fluctuation state 2. From the perspective of overall efficiency, we can see that LSTM-HMM(q) and LSTM-HMM(m) have more accurate predictions. Based on the confusion matrixes shown in Fig 9, we calculated the accuracy vector necessary for the evaluation and the details are shown in Table 12.

As is shown in Table 12, the LSTM-HMM(m) has the highest accuracy rate, which is 0.71, with the highest kappa of 0.58 and the highest AUC of 0.79, showing a moderate consistency with the real values and a good performance in prediction. The LSTM-HMM(m) also has good performance in prediction, with an accuracy rate of 0.70. While the GMM-HMM(q), HMM(q) and HMM(m) have a lower prediction accuracy, kappa and AUC. When the model input transforms from quarterly to monthly, the model performance of the HMM and GMM-HMM can be improved in terms of accuracy, kappa and AUC, while the LSTM-HMM, with either monthly or quarterly input, have similar model performance.

To sum up, within the 6-year time window which involves 24 observations for each round of training, the LSTM-HMM(q) with monthly input has the best performance.

**Models within 8-year time window.**    We run the six models including HMM(q), GMM-HMM(q), LSTM-HMM(q), HMM(m), GMM-HMM(m) and LSTM-HMM(m) within 8-year time windows and their confusion matrixes are shown in Fig 10.

As we can see from the Fig 10, there are 48 predictions in total, and the characteristics of these confusion matrixes are similar with that of models within 4-year time window and 6-year time window. All these models have accurate predictions of GDP fluctuation states of value 0. The LSTM-HMM with monthly input has the best performance since it has 35 precise predictions among the 48 predictions. Based on the confusion matrixes shown in Fig 10, we calculated the accuracy vector necessary for the evaluation in Table 13.

As is shown in Table 13, LSTM-HMM(m) with monthly input has the highest prediction accuracy, which is 0.73, with the highest kappa of 0.58 and the highest AUC of 0.80. LSTM-HMM(m) has a moderate consistency with the real values and a better performance in prediction than that of HMM(q). HMM(m), GMM-HMM(q) and GMM-HMM(m). LST<-HMM(q) also have good performances in prediction with an accuracy rate of over 0.71. When the model input transforms from quarterly to monthly, the model performance of the HMM, GMM-HMM and LSTM-HMM can be improved in terms of accuracy, kappa and AUC.

To sum up, within the 8-year time window which involves 32 observations for each round of training, the LSTM-HMM(m) with monthly univariate-input has the best performance, while HMM(q) and GMM-HMM(m) also perform well in prediction.

**Models within 10-year time window.**    We run the six models including HMM(q), GMM-HMM(q), LSTM-HMM(q), HMM(m), GMM-HMM(m) and LSTM-HMM(m) within 10-year time windows and their confusion matrixes are shown in Fig 11.

As we can see from the Fig 11, there are 40 predictions in total, and the characteristics of these confusion matrixes are different with that of models in the above subsections. All these models have relatively precise predictions, where there are many predictions of GDP fluctuation state 0 but a smaller number of predictions of GDP fluctuation state 2.. These predictions could be attributed to the selection bias of training data, since there are much more GDP fluctuation states with the value 0 within the 10-year window rolling from 2010, where the historical GDP fluctuation states of value 0 may have an excessive influence on the forecast models and result more predictions of value 0. From the perspective of overall efficiency, LSTM-HMM has more precise predictions. We then try to evaluate them according to the accuracy parameters shown in Table 14.

As is shown in Table 14, the LSTM-HMM with monthly univariate-input has the highest accuracy rate, which is 0.80, and the highest AUC of 0.85, while the kappa is 0.67, which shows that the predictions of this model have substantial consistency. When the model input transforms from quarterly to monthly, the model performance of the HMM, GMM-HMM and LSTM-HMM keep the same or have small improvement in terms of accuracy, kappa and AUC.

To sum up, within the 10-year time window which involves 40 observations for each round of training and 40 predictions, the LSTM-HMM with monthly univariate input has the best performance in prediction, while the other models also perform well in the prediction of GDP fluctuation states.

**Comparing models within different time windows.**    We compare their ROC curves within different time windows, which are shown below in Fig 12.

For 4-year time window in Fig 12(a), there is a spread between ROC curves, where GMM-HMM(m), LSTM-HMM(q) and LSTM-HMM(m) performs better than the other models. For 6-year time window in Fig 12(b), the ROC curves are similar with the ROC curves of

4-year time window, while there is a larger spread between the ROC of GMM-HMM(q) and that of others. This may result from the CPI classifier and the quarterly input, as GMM-HMM with monthly input and HMM with quarterly input have better performance in terms of ROC. For 8-year time window in Fig 12(c), there is a clear spread between the ROC curves of LSTM-HMM and the ROC curves of the other models, which shows an improved model performance with the proposed LSTM-HMM. For 10-year time window in Fig 12(d), ROC curves share similar trends with a even larger spread between the ROC curves of LSTM-HMM and that of others. We can see from Fig 12 that, within all the time windows, LSTM-HMM(m) and LSTM-HMM(q) generally have better performances in prediction than the other models. The ROC curves of different models share similar trends within each time window.

We also compare their accuracy vector: (accuracy, kappa, AUC) shown in Table 15, where we find that from the perspective of time windows, we find that models within 10-year time windows generally have higher prediction accuracy and higher consistency in predictions, and all the models also have good performances in predictions within 10-year time windows; from the perspective of models, we find that LSTM-HMM structures generally have good prediction accuracy; from the perspective of overall forecasting effect, LSTM-HMM with either quarterly or monthly input within 10-year time windows has the best performance with the highest prediction accuracy and consistency; from the perspective of model input, when the model input transforms from quarterly to monthly, the model performance of the HMM, GMM-HMM and LSTM-HMM can generally be improved in terms of accuracy, kappa and AUC.

There are several reasons for these model results: firstly, 10-year window with 40 observations in training is long enough for us to observe all types of GDP fluctuation states while avoiding the bias caused by too many observations of certain type as a proper data source for LSTM-HMM system; secondly, LSTM-HMM take more effects of historical CPI into the training process with the application of LSTM and tend to predict a suitable real time CPI fluctuation state, which replaces the lagged observable states adopted by other models, and helps in the prediction of GDP fluctuation using HMM; thirdly, since monthly data in the same season is more informative than quarterly data observed in the past season, the models with monthly input performs better than the models with quarterly input in most conditions.

## Conclusion

This paper establishes a Long Short Term Memory Recurrent Neural Network-Hidden Markov Model (LSTM-HMM) to generate real time CPI fluctuation state and based on which predict hidden GDP fluctuation state. We compare the predictive power of this model with other dynamic forecast systems of China's quarterly GDP fluctuation state, including HMM, GMM-HMM, and LSTM-HMM with an input of monthly CPI or quarterly CPI within 4-year, 6-year, 8-year and 10-year time window. The LSTM-HMM built in this paper generally performs well in prediction and is testified to be an efficient forecast model of GDP fluctuation states especially when using monthly input that conveys more historical information. The detailed conclusions of the comparison of models are as follows.

Firstly, we find that with an input of quarterly CPI, LSTM-HMM performs better within each time window. Secondly, we find that with an input of monthly CPI, GMM-HMM performs better within 4-year time window and LSTM-HMM performs better within 8-year time window, 8-year time window 10-year time window. Thirdly, among all the time windows, models within 8-year time window and 10-year time window have better overall performance in accuracy and consistency, which could avoid certain bias caused by too many observations of single type. HMM employed in our analyses depends on parameters calculated from sample data within certain time period, which means that these parameters change with time window

and build different models. In the empirical analysis of GDP fluctuation states forecast, models encapsulated with HMM tend to ignore the effect of the sparse states and the giant difference of numbers of states will result in selection bias. In this paper, we reduce the impact of selection bias by experiments within pre-selected lengths of time window ranging from 4-year, 6-year, 8-year to 10-year and find that within 10-year window, the model performance is better. Fourthly, LSTM-HMM generally has the best accuracy and consistency, since this model predicts the real time CPI fluctuation states and take it as the observable state using LSTM, making the most of information conveyed by the monthly input and detecting useful historical information within a suitable time window.

## Supporting information

**S1 Appendix. In the appendix, we give a brief explanation of Viterbi algorithm involved in this paper referring to [22], and take the decoding process of HMM in the benchmark model: HMM as an example of using Viterbi algorithm, where we find the most likely hidden state sequence that explains the observations.**
(PDF)

## Author Contributions

**Conceptualization:** Junhuan Zhang.

**Data curation:** Jiaqi Wen, Zhen Yang.

**Formal analysis:** Junhuan Zhang, Jiaqi Wen, Zhen Yang.

**Funding acquisition:** Junhuan Zhang.

**Methodology:** Junhuan Zhang.

**Software:** Jiaqi Wen, Zhen Yang.

**Supervision:** Junhuan Zhang.

**Writing – original draft:** Jiaqi Wen.

**Writing – review & editing:** Junhuan Zhang.

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
