## [Decision Letter · Decision Letter 0]

12 Jul 2021

PONE-D-21-12122

China's GDP Forecasting using Long Short Term Memory Recurrent Neural Network and Hidden Markov Model

PLOS ONE

Dear Dr. Zhang,

Thank you for submitting your manuscript to PLOS ONE. After careful consideration, we feel that it has merit but does not fully meet PLOS ONE’s publication criteria as it currently stands. Therefore, we invite you to submit a revised version of the manuscript that addresses the points raised during the review process.

We look forward to receiving your revised manuscript.

Kind regards,

José Soares Andrade Jr.

Academic Editor

PLOS ONE

Journal Requirements:

Reviewers' comments:

Reviewer's Responses to Questions

**Comments to the Author**

1. Is the manuscript technically sound, and do the data support the conclusions?

Reviewer #1: Partly

Reviewer #2: Yes

2. Has the statistical analysis been performed appropriately and rigorously? 

Reviewer #1: I Don't Know

Reviewer #2: Yes

3. Have the authors made all data underlying the findings in their manuscript fully available?

Reviewer #1: No

Reviewer #2: Yes

4. Is the manuscript presented in an intelligible fashion and written in standard English?

Reviewer #1: Yes

Reviewer #2: Yes

5. Review Comments to the Author

Reviewer #1: The authors propose a Long Short Term Memory Recurrent Neural Network and Hidden Markov Model, a forecasting model to predict the China GDP using as input the CPI.  They compare the performance of their algorithm with different but still similar techniques. The conclusion is somehow not very clear, depending on the time window, one can have a different rank of accuracy for the models. However, the authors claim that under some conditions the LSTM-HMM outperforms the other models. 

In its current form, the paper reads like a contribution to a specialized journal or workshop about forecasting models. Yet, I believe a Plos One paper should appeal to a wider audience.

1 -  The authors should consider reducing the number of abbreviations in the abstract, and properly define some of them along with the abstract (and article), such as GDP and CPI. 

2 -  The innovation point of the article is not clear enough. In the abstract and in the discussion it is not very clear what are the findings, and more important why might that be relevant? 

3 - What is the motivation to introduce a LSTM? 

4 - In section 2.1, I found the notation confusing. The authors write that "$S$ is a discrete set (...), where $t$ stands for time.", but  I do not see $t$ before that. After this phrase, the authors start to use $s_t$, is this the same capital $S$ defined before?

5 - In Section 3.1, when describing Fig. 5 the authors mention "the curve" and "the straight line". I suppose this is a typo. 

6 - In section 3.2, the Granger does not exactly measure causality, therefore it is better to use the term Granger predictive causality, and clarify this in the text. 

7 - The results of the article are based on the comparison of numbers without any clear interpretation. Many of the numerical results have four to five decimal digits (e.g. $gamma$). Is this precision really significant? For instance, the authors draw conclusions and compare models with an accuracy of 0.6406 and 0.6563. Is this difference really significant?

Reviewer #2: I have studied the manuscript “China's GDP Forecasting using Long Short Term Memory Recurrent Neural Network

and Hidden Markov Model” by Junhuan Zhang.

Here the author consider the framework of Depp Learning algorithms in order to predict the growth rate of China’s GDP. Precisely, the author compare the predictive

power of the Long Short Term Memory Recurrent Neural Network (LSTM-HMM) with other dynamic forecast systems. From the results shown, taking into account

quarterly and monthly inputs of CPI, the performance of the algorithms analysed depends on the time window considered. For quarterly input:

i) with an input of quarterly CPI, LSTM-HMM performs better within four-year time window and six-year time window;

ii) HMM performs better within eight-year time window;

ii) GMM-HMM performs better within ten-year time window;

And for monthly input:

i) GMM-HMM performs better within four-year time window and six-year time window;

ii) LSTM-HMM performs better within eight-year time window;

iii) HMM performs better within ten-year time window;

Moreover, among all the time windows, models within eight-year time window have better overall performance in accuracy and consistency and LSTM-HMM with

an input of monthly CPI generally has good precision, and within eight-year time window it has the best accuracy and consistency.

Considering previous numerical studies and my understanding of results, the findings of this manuscript are predictable and lack novelty. The authors just observed finite-size effects when considered network aspect ratios and boundary conditions (weighted by the electrode dimension) on the convergence point and span of the emergent region. However, the manuscript is scientifically valid and considering the Editorial Criteria, it is publishable. But there are several points that authors should clarify and make up for deficiencies at first.

My comments/questions are as follows:

1. Throughout the text the author describes several properties without specifying what CPI means. Therefore, the author must need to define and make clear what CPI means.

2. In the last paragraph of the Conclusion Section the author states that "In practical application, we should try our best to reduce the impact of selection bias.". The author should clarify what he means by "practical application" and what kinds of selection bias is he talking about.

4. Considering the results of the ROC curves, there is not a monotonic behavior in performance of the algorithms with the time window. Especially for eight-year time window, theres is a clear spread in the ROC curves. I think that the author must provide some indication as to why this is so.

3. The author must be revise all figures and tables of the manuscript, with clear description in the respective captions, in order to improve the quality and readability of the manuscript. For instance, there is a typo in Table 13.

6. PLOS authors have the option to publish the peer review history of their article (what does this mean?). If published, this will include your full peer review and any attached files.

Reviewer #1: No

Reviewer #2: No

---

## [Author Response · Author response to Decision Letter 0]

21 Sep 2021

Authors thank reviewers for their helpful and valuable comments. Please find the attached response to reviewers in the revised manuscript.

---

## [Decision Letter · Decision Letter 1]

25 Jan 2022

PONE-D-21-12122R1China's GDP Forecasting using Long Short Term Memory Recurrent Neural Network and Hidden Markov ModelPLOS ONE

Dear Dr. Zhang,

Thank you for submitting your manuscript to PLOS ONE. After careful consideration, we feel that it has merit but does not fully meet PLOS ONE’s publication criteria as it currently stands. Therefore, we invite you to submit a revised version of the manuscript that addresses the points raised during the review process.

 Your manuscript has been reviewed by two of our reviewers. Another reviewer was consulted, but we have been informed that no report will be received. In view of the assessment from Reviewer#3, we believe that a minor revision is still necessary before we can consider the manuscript further.

We look forward to receiving your revised manuscript.

Kind regards,

José S. Andrade Jr.

Academic Editor

PLOS ONE

Reviewers' comments:

Reviewer's Responses to Questions

**Comments to the Author**

1. If the authors have adequately addressed your comments raised in a previous round of review and you feel that this manuscript is now acceptable for publication, you may indicate that here to bypass the “Comments to the Author” section, enter your conflict of interest statement in the “Confidential to Editor” section, and submit your "Accept" recommendation.

Reviewer #1: All comments have been addressed

Reviewer #3: (No Response)

2. Is the manuscript technically sound, and do the data support the conclusions?

Reviewer #1: Yes

Reviewer #3: Partly

3. Has the statistical analysis been performed appropriately and rigorously? 

Reviewer #1: I Don't Know

Reviewer #3: Yes

4. Have the authors made all data underlying the findings in their manuscript fully available?

Reviewer #1: Yes

Reviewer #3: (No Response)

5. Is the manuscript presented in an intelligible fashion and written in standard English?

Reviewer #1: Yes

Reviewer #3: Yes

6. Review Comments to the Author

Reviewer #1: The authors have adequately addressed my comments __________________________________________________

Reviewer #3: The authors present a Hidden Markov Model coupled with an Artificial Recurrent Neural Network Architecture to make predictions on a discrete time series constructed using China's GDP fluctuation data. As input to the model, they use discrete data constructed using time series from China's CPI fluctuation and compare the results of their model with some other forecasting models using different time windows. The authors claim that their model performs better in an eight-year time window, showing good precision and better accuracy when compared to the other models tested.

Overall, the manuscript is fairly well written, however the labels within the figures are very small. The paper is quite long with many small results that make it difficult for the reader to understand which are the most important to support the author's main claims. In order to improve in this aspect, I suggest that the main results should be highlighted for better readability. Despite this, the manuscript presents good results and an interesting approach that connects concepts of Econometrics and Deep Learning. For these reasons it seems appropriate for Plos One, however I have a few questions I would like addressed:

1. In section 3.2, the authors write: "...which shows that under the significance level of 10 percent, we cannot reject the hypothesis of no cointegration relationship since the statistical value is 29.83, higher than the critical value of 27.43 at right tail.". How do the authors think that cointegration may, or may not, affect the results found for their model?

2. In section 3.4, the authors define how the time series of GDP fluctuation will be discretized. This is one of the most important parts of the manuscript as it can affect the quality of the time series. Figures 8, 9 and 10 clearly show the confusion matrices generated from an unbalanced test set with many observations of 0's. Are training sets suffering from the same problem? what are the impacts of choosing different dividing points on the quality of the training set?

3. Still in section 3.4, the authors support the choice of division points based on a 2008 paper that uses data from 1990 to 2006 to define the turning points of the GDP growth rate curve. In this sense, in some time windows shown in Table 9, the models are using information from the future in their predictions. Have the authors tested adapting the choice of dividing points to time windows to avoid this kind of inconsistency?

7. PLOS authors have the option to publish the peer review history of their article (what does this mean?). If published, this will include your full peer review and any attached files.

Reviewer #1: No

Reviewer #3: No

---

## [Author Response · Author response to Decision Letter 1]

1 Mar 2022

Please check the attached response letter.

---

## [Decision Letter · Decision Letter 2]

24 May 2022

China's GDP Forecasting using Long Short Term Memory Recurrent Neural Network and Hidden Markov Model

PONE-D-21-12122R2

Dear Dr. Zhang,

We’re pleased to inform you that your manuscript has been judged scientifically suitable for publication and will be formally accepted for publication once it meets all outstanding technical requirements.

Kind regards,

José S. Andrade Jr.

Academic Editor

PLOS ONE

Additional Editor Comments (optional):

Reviewers' comments:

Reviewer's Responses to Questions

**Comments to the Author**

1. If the authors have adequately addressed your comments raised in a previous round of review and you feel that this manuscript is now acceptable for publication, you may indicate that here to bypass the “Comments to the Author” section, enter your conflict of interest statement in the “Confidential to Editor” section, and submit your "Accept" recommendation.

Reviewer #3: All comments have been addressed

2. Is the manuscript technically sound, and do the data support the conclusions?

Reviewer #3: Yes

3. Has the statistical analysis been performed appropriately and rigorously? 

Reviewer #3: Yes

4. Have the authors made all data underlying the findings in their manuscript fully available?

Reviewer #3: Yes

5. Is the manuscript presented in an intelligible fashion and written in standard English?

Reviewer #3: Yes

6. Review Comments to the Author

Reviewer #3: (No Response)

7. PLOS authors have the option to publish the peer review history of their article (what does this mean?). If published, this will include your full peer review and any attached files.

Reviewer #3: No

---

## [Editor Report · Acceptance letter]

3 Jun 2022

PONE-D-21-12122R2 

China’s GDP Forecasting using Long Short Term Memory Recurrent Neural Network and Hidden Markov Model 

Dear Dr. Zhang:

I'm pleased to inform you that your manuscript has been deemed suitable for publication in PLOS ONE. Congratulations! Your manuscript is now with our production department. 

Kind regards, 

on behalf of

Prof José S. Andrade Jr. 

Academic Editor

PLOS ONE